# Comparative efficacy and acceptability of interventions for insomnia in breast cancer patients: A protocol for systematic review and network meta-analysis

Zhifan Li[1], Qian Wang[1], Junxia Xu[1], Qihua Song[1], Xiaoling Ling[2], Ya Gao[3,4], Junqiang Lei[1,5]*

1 The First Clinical Medical College of Lanzhou University, Lanzhou, China, 2 Department of Oncology, The First Hospital of Lanzhou University, Lanzhou, China, 3 Department of Health Research Methods, Evidence, and Impact, McMaster University, Hamilton, ON, Canada, 4 Evidence-Based Medicine Center, School of Basic Medical Sciences, Lanzhou University, Lanzhou, China, 5 Department of Radiology, The First Hospital of Lanzhou University, Lanzhou, China

* leijq2011@126.com

**Data Availability Statement:** No datasets were generated or analysed during the current study.

## Abstract

### Background

Symptoms of insomnia are highly prevalent in patients with breast cancer. There are a large number of pharmacological and non-pharmacological interventions that can be used for the management of insomnia in breast cancer patients; however, their comparative effectiveness and acceptability remain uncertain. This review aims to evaluate the efficacy and acceptability of different interventions for insomnia in breast cancer patients using a Bayesian network meta-analysis (NMA).

### Methods

We will perform a comprehensive literature search in PubMed, EMBASE, Cochrane Central Register of Controlled Trials (CENTRAL), Web of Science, and PsycINFO from inception to November 2022. We will include randomized controlled trials (RCTs) that compared the effects of different interventions on the management of insomnia in breast cancer patients. We will assess the risk of bias assessment using a modified Cochrane instrument. We will conduct a Bayesian random-effects framework NMA to estimate relative effects of interventional procedures. We will use Grading of Recommendations Assessment, Development and Evaluation to rate the certainty of evidence.

### Discussion

To our knowledge, this will be the first systematic review and network meta-analysis to compare the effectiveness and acceptability of all currently available interventions for insomnia in patients with breast cancer. The results of our review will help provide more evidence for the treatment of insomnia in breast cancer patients.

**Funding:** The authors received no specific funding for this work.

**Competing interests:** The authors have declared that no competing interests exist.

**Abbreviations:** CBT-I, cognitive-behavioral therapy for insomnia; NMA, network meta-analysis; RCTs, randomized controlled trials; SE, sleep efficiency; TST, total sleep time; SOL, sleep onset latency; WASO, wake after sleep onset; ISI, insomnia Severity Index; PSQI, Pittsburgh Sleep Quality Index.

## Systematic review registration

PROSPERO registration number CRD42021282211.

## Introduction

Insomnia is the most common sleep-wake disorder in cancer patients [1], especially in patients with breast cancer, lung cancer, head, and neck cancer [2, 3]. The prevalence of insomnia in cancer patients can reach 30% to 60%, which is almost 2 to 3 times that of the general population [4–7]. Factors contributing to the high incidence of insomnia in cancer populations include the psychological response to various stressors (such as cancer diagnosis) experienced, the direct impact of cancer treatment, and its side effects [8]. Compared with other cancer sites (such as prostate, gynecology, head and neck, or urinary), patients with breast cancer have the highest prevalence of insomnia (40%-70%) [5, 9, 10]. For breast cancer patients, in addition to the stress caused by the cancer diagnosis, some treatment-related factors are often related to insomnia. Vasomotor symptoms, including hot flashes and night sweats, are common with chemotherapy-induced amenorrhea and endocrine therapies and can cause insomnia episodes or aggravate pre-existing conditions [11–14]. Insomnia not only harms physical and social functions, but may also cause depression and anxiety [8], and may even increase the fatality rate of breast cancer patients [15–17].

Treatment options for insomnia include pharmacological and non-pharmacological interventions. The current international guidelines [18, 19] only recommend drugs for short-term treatment of insomnia (≤4 weeks), because although the drug is effective in the short term, it has negative side effects and major risks, and long-term high-dose treatment is not recommended [20]. Cognitive-behavioral therapy for insomnia (CBT-I) is considered the gold standard non-pharmacological treatment for insomnia [21]. However, despite the evidence that CBT-I is effective, CBT-I has not been widely used due to limited access to well-trained providers and poor adherence to the recommended treatment [22]. In addition to CBT-I, other non-pharmacological treatment options are increasingly used in the treatment of insomnia. As a relatively safe intervention, acupuncture is now widely accepted and used, and may be a promising treatment option. Some systematic reviews revealed that symptoms of insomnia in patients treated with acupuncture may be significantly improved compared with pharmacological treatments [23–26]. Some studies have also shown that mindfulness meditation can significantly reduce the severity of insomnia, prolong the total sleep time, and improve the sleep efficiency and quality of patients with chronic insomnia [27–30]. Yoga is a novel and effective intervention to improve the symptoms of insomnia and may become a new addition to the insomnia management program [31–33]. In addition, music therapy [34] and Tai Chi [35, 36] have also been shown to help improve patients' sleep quality.

Although there is one systematic review [37] that evaluated the relative efficacy and safety of pharmacological and non-pharmacological interventions, this study only focused on elderly patients with insomnia rather than breast cancer patients. Therefore, we propose to conduct a systematic review and network meta-analysis (NMA) of randomized controlled trials (RCTs) to evaluate the relative effectiveness and acceptability of existing interventions in the treatment of insomnia in breast cancer patients.

## Methods

Our protocol follows the Preferred Reporting Items for Systematic Reviews and Meta-Analyses for Protocols (PRISMA-P) guideline [38]. We have registered the protocol on the international prospective register of systematic review (PROSPERO) (CRD42021282211).

### Information sources and search strategy

We will search PubMed, EMBASE, Cochrane Central Register of Controlled Trials (CENTRAL), Web of Science, and PsycINFO from inception to November 2022. Search terms will include controlled vocabulary and free text synonyms comprising randomized controlled trials, breast neoplasm, and insomnia. There will be no restriction on the language of publication. For publications that are not in English and Chinese, we will seek translation assistance. The search strategy for each database is provided in detail in S1 Appendix. The reference lists of relevant systematic reviews/meta-analyses and included RCTs will be checked to identify additional potentially eligible studies.

### Eligibility criteria

#### Types of studies

Randomized controlled trials comparing the effects of different interventions on the management of insomnia in breast cancer patients will be included. RCTs should report at least one measure outcome of sleep and provide enough detail to calculate effect sizes. We will exclude cluster or crossover RCTs.

#### Population

Individuals with a breast cancer diagnosis who had clinically relevant levels of insomnia as defined by a standardized diagnostic system such as the Diagnostic and Statistical Manual of Mental Disorders (DSM) [39], International Classification of Sleep Disorders (ICSD) [40], Tenth revision of the International Statistical Classification of Diseases and Health-Related Problems (ICD-10) [41], and other well-recognized classifications. We will put no restrictions on the subtype, histology, stage, and treatment of breast cancer, age and comorbidities of patients, and duration of insomnia.

#### Intervention

Any pharmacological and non-pharmacological interventions used for the management of insomnia in breast cancer patients. The eligible interventions are listed in Table 1.

#### Comparator

The controls will include waiting list, treatment/care as usual, placebo, or a different pharmacological or non-pharmacological intervention. We will investigate the potential clinical and statistical heterogeneity of non-interventional controls (waiting list, treatment/care as usual, placebo) to determine whether to treat them as a single node.

#### Outcome

Outcomes of interest will include sleep efficiency (SE, %), total sleep time (TST), sleep onset latency (SOL), wake after sleep onset (WASO), and acceptability. SE, SOL, WASO, and TST are typically defined as a ratio of time asleep to time in bed, the length of time after lights out until sleep onset, the length of waking time after the onset of persistent sleep, and total

**Table 1. Eligible interventions.**

| Intervention | |
| --- | --- |
| **Pharmacotherapy** | Benzodiazepines: short, medium, and long-acting substances e.g. triazolam, lormetazepam, flurazepam |
| | Benzodiazepine-receptor-agonists ("Z Drugs"): zopiclone, eszopiclone, zolpideme, zaleplon |
| | Sedating antidepressants: e.g. trimipramin, doxepine, amitryptilin, opipramol |
| | Sedating antipsychotics: e.g. promethazine, chlorprothixene, quetiapine, olanzapine or risperidone |
| | Melatonin or melatonin- receptor-agonists: ramelteon, agomelatonin, tasimelteon |
| | Herbal preparations: e.g. valerian, chamomile, hops, kava-kava, passionflower, St. John's wort, oat |
| | Nutrients: e.g. L-tryptophan, magnesium |
| | Traditional Chinese medicine |
| | Others |
| **CBT-I and related approaches** | Psychoeducation |
| | Sleep hygiene methods |
| | Stimulus control |
| | Sleep restriction or compression |
| | Cognitive therapy |
| | Problem-solving strategies |
| | Paradoxical intention |
| | Relaxation methods |
| | Others |
| **Other approaches** | Bright-light therapy |
| | Exercise intervention |
| | Electric stimulation techniques |
| | Transcranial magnetic stimulation techniques |
| | Hypnosis |
| | Music-therapy |
| | Yoga |
| | Tai Chi |
| | Acupuncture |
| | Others |

nighttime sleep, respectively. In addition, the scores from the Insomnia Severity Index (ISI) or the Pittsburgh Sleep Quality Index (PSQI) are also the outcomes that we are interested in. For both questionnaires, a higher score indicates more severe insomnia (ISI) or worse sleep quality (PSQI). To measure the acceptability of treatments, we will use the proportion of patients who stopped treatment for any reason (all-cause discontinuation) [42]. Acceptability reflects how well the intervention was accepted by patients and whether they continued to use the intervention because it was effective.

## Study selection and screening

We will import the retrieved records into EndNote X9 (Thomson Reuters (Scientific) LLC Philadelphia, PA, US) software for management. Pairs of reviewers will screen out the potentially relevant studies by reading the titles and abstracts of all articles. The full-text review will be performed independently by the same review authors according to the inclusion and

exclusion criteria. Disagreements will be resolved by discussion. If the dispute cannot be resolved by discussion, we will consult a third review author.

## Data extraction

Pairs of reviewers will independently extract the data using a standardized form, which covers the following items: (1) basic characteristics, including first author, publication year, country, funding, and study design; (2) patient characteristics, including grouping and sample size, age and sex of patients, cancer stage and type, diagnostic methods of insomnia, insomnia duration; (3) intervention characteristics, including experimental and control methods, treatment duration, and length of follow-up; (4) outcomes of interest. We will solve disagreements through discussion with the third reviewer.

## Risk of bias assessment

We will assess the risk of bias of included randomized controlled trials using a revised version of the Cochrane tool for assessing the risk of bias in randomized trials (RoB 2.0) [43, 44]. The assessment list includes the following domains: bias arising from the randomization process, bias due to deviations from intended interventions, bias due to missing outcome data, bias in the measurement of the outcome, and bias in the selection of the reported result. We will rate each domain as either: low risk of bias, some concerns—probably low risk of bias, some concerns—probably high risk of bias, or high risk of bias. We will rate trials at high risk of bias overall if one or more domains are rated as some concerns—probably high risk of bias, or as high risk of bias and as low risk of bias overall if all domains are rated as some concerns—probably low risk of bias or low risk of bias. In our study, one reviewer will rate the risk of bias of each study according to the scale and another will review it. Reviewers will resolve discrepancies by discussion and, when not possible, with adjudication by a third party.

## Data synthesis

We will conduct pairwise meta-analyses using a Bayesian framework with the random-effects model. We will use odds ratio (OR) with 95% credible intervals for the dichotomous variable (ie. acceptability). For continuous variables, we will use standardized mean differences or mean differences with 95% credible intervals. The heterogeneity between head-to-head trials will be estimated using $I^2$ statistics and $P$-value. The values of 25%, 50%, and 75% for the $I^2$ as indicative of low, moderate, and high statistical heterogeneity, respectively. We will evaluate the publication bias with the Egger test and funnel plots if the number of studies exceeded 10 [45]. A 2-tailed $P$ value $< 0.05$ is considered statistically significant.

For network meta-analysis, we will create a network plot to present the geometry of the network of comparisons across trials using Stata version 15.1 (StataCorp, College Station, TX). The NMA will be conducted on both direct evidence and indirect evidence in a Bayesian random-effects framework using gemtc R package in R version 3.6.3 (RStudio, Boston, MA) [43]. For all analyses, we will use three chains with 100 000 iterations after an initial burn-in of 10 000 and a thinning of 10. We will use the deviance information criterion (DIC) to compare model fit and parsimony. The convergence will be assessed using the Brooks-Gelman-Rubin (BGR) plots method [46]. We will assess the global heterogeneity for all comparisons from the network meta-analysis models using the $I^2$ statistic with the gemtc R package. We will use the node-splitting method to examine the inconsistency between direct and indirect estimates [47]. We will evaluate, according to the surface under the cumulative ranking curve, the rank probabilities for interventions [48]. We will generate comparison-adjusted funnel plots to explore the presence of small sample effects among the networks [49].

## Subgroup analysis and sensitivity analysis

Where sufficient data are available, we will conduct subgroup analyses or meta-regression analyses based on country and mean age of patients; subtype, histology, stage, and treatment (eg, surgery, chemotherapy, endocrine therapy) of breast cancer; comorbid disorders, and duration of interventions and insomnia. We will also conduct sensitivity analyses to explore the influence of variables on the outcomes. Planned sensitivity analyses will exclude trials with a high risk of bias or significant levels of missing data.

## Grading the strength of evidence

We will assess the certainty of the direct, indirect, and network estimate for all outcomes according to the Grading of Recommendations Assessment, Development and Evaluation (GRADE) approach, considering the risk of bias, inconsistency, indirectness, publication bias, intransitivity, incoherence (difference between direct and indirect effects), and imprecision [50, 51]. We will rate the certainty for each comparison and outcome as high, moderate, low, or very low. Two reviewers with experience in using GRADE will rate each domain for each comparison and outcome separately, and resolve differences through discussion.

## Discussion

Insomnia has become a prevalent and significant public health problem [52]. It is common both in the general population and in all cancer patients, but has been shown to be particularly prevalent among breast cancer patients [53, 54]. Approximately 30%-50% of breast cancer survivors have insomnia persisting years after treatment [53]. Stress, anxiety, discomfort after surgery such as chronic pain after breast surgery, and side effects of treatment such as hot flashes and night sweats, which are common after chemotherapy and endocrine therapy, can all contribute to insomnia in breast cancer patients [12, 14, 53, 55, 56]. Insomnia may be related to the occurrence of some adverse health outcomes, such as poor physical health, poor mental health, including symptoms of anxiety and depression, and decreased quality of life [57, 58]. Therefore, there is an urgent need to explore a safe, efficient, and easily available treatment method for insomnia in breast cancer patients. So far, many studies have been conducted on insomnia in breast cancer patients. The therapeutic effects of CBT-I, acupuncture, Tai Chi, and mindfulness-based interventions on insomnia in breast cancer patients have been proven by multiple randomized controlled trials [59–64]. There is also a systematic review that evaluated the therapeutic effect of CBT-I [65]. However, there is still a lack of high-quality review of evidence summarizing the effectiveness and harm of different interventions for insomnia in breast cancer patients.

Our research will explore the comparative effectiveness and acceptability of all currently available interventions for insomnia in patients with breast cancer, and we will use the GRADE approach to assess the certainty of the evidence supporting the effect of treatment. The state-of-the-art methodology will also be used to summarize the relative effectiveness of competing interventions. The results of our review will help provide more evidence for the treatment of insomnia in breast cancer patients.

## Supporting information

**S1 Appendix. Search strategy for each database.**
(DOCX)

**S1 Checklist. PRISMA-P (Preferred Reporting Items for Systematic review and Meta-Analysis Protocols) 2015 checklist: Recommended items to address in a systematic review**

protocol*.
(DOC)

## Author Contributions

**Conceptualization:** Zhifan Li, Qian Wang, Junxia Xu, Qihua Song, Xiaoling Ling.

**Investigation:** Zhifan Li, Qian Wang.

**Methodology:** Zhifan Li, Xiaoling Ling, Ya Gao, Junqiang Lei.

**Project administration:** Junqiang Lei.

**Resources:** Zhifan Li, Qian Wang, Junxia Xu, Xiaoling Ling.

**Supervision:** Junqiang Lei.

**Visualization:** Zhifan Li, Qian Wang, Junxia Xu, Qihua Song.

**Writing – original draft:** Zhifan Li.

**Writing – review & editing:** Zhifan Li, Ya Gao, Junqiang Lei.

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
