## [Decision Letter · Decision Letter 0]

28 Oct 2022

PONE-D-22-24531Comparative efficacy and acceptability of interventions for insomnia in breast cancer patients: A protocol for systematic review and network meta-analysisPLOS ONE

Dear Dr. Jun-Qiang Lei

Thank you for submitting your manuscript to PLOS ONE. After careful consideration, we feel that it has merit but does not fully meet PLOS ONE’s publication criteria as it currently stands. Therefore, we invite you to submit a revised version of the manuscript that addresses the points raised during the review process.

Dear authors,

the topic of the present article titled “Comparative efficacy and acceptability of interventions for insomnia in breast cancer patients: A protocol for systematic review and network meta-analysis” is very interesting, the paper and the aim falls within the scope of the journal but the article needs major improvements.

Review some spellings. Methods and results section and discussion should be modified and improved.

I suggest improving the manuscript with the reviewers' comments

We look forward to receiving your revised manuscript.

Kind regards,

Andrea Giannini

Academic Editor

PLOS ONE

Journal Requirements:

Reviewers' comments:

Reviewer's Responses to Questions

**Comments to the Author**

1. Does the manuscript provide a valid rationale for the proposed study, with clearly identified and justified research questions?

Reviewer #1: Yes

Reviewer #2: Yes

Reviewer #3: No

Reviewer #4: Partly

2. Is the protocol technically sound and planned in a manner that will lead to a meaningful outcome and allow testing the stated hypotheses?

Reviewer #1: Yes

Reviewer #2: Yes

Reviewer #3: Yes

Reviewer #4: Partly

3. Is the methodology feasible and described in sufficient detail to allow the work to be replicable?

Reviewer #1: Yes

Reviewer #2: Yes

Reviewer #3: Yes

Reviewer #4: Yes

4. Have the authors described where all data underlying the findings will be made available when the study is complete?

Reviewer #1: Yes

Reviewer #2: No

Reviewer #3: Yes

Reviewer #4: Yes

5. Is the manuscript presented in an intelligible fashion and written in standard English?

Reviewer #1: Yes

Reviewer #2: Yes

Reviewer #3: Yes

Reviewer #4: Yes

6. Review Comments to the Author

You may also provide optional suggestions and comments to authors that they might find helpful in planning their study.

Reviewer #1: I think the article is very interesting and current. From our knowledge we know that insomnia is determined not only by the diagnosis of the tumor but also by its treatment. it happens very often that after an almost complete recovery, hormone therapy exacerbates a previous insomnia or creates alterations in the sleep wake cycle de novo. I believe it is essential to stress this speech in the introduction and discussion.

Reviewer #2: Question 4 - The authors have stated that all relevant data from the study will be made available upon study completion; this is a protocol, not a completed study.

- The study will explore the "comparative efficacy and acceptability" of interventions for insomnia in breast cancer patients. Efficacy of interventions will be able to be shown once the data is analyzed; however, what is meant by "acceptability" and how will that be explored and analyzed?

- It is surprising that there will be "...no restriction on the language of publication" of studies to be selected. Will there be translation help furnished, or are the authors knowledgable in a large number of languages?

- The controls will be "...waiting list, treatment/care as usual, etc." Do the authors mean to assemble control patients, find controls outside of the selected articles, or use the "controls" who are presented in the RCTs used?

- It is surprising that, out of seven authors, only two will screen all articles and review all the selected articles. Are the remaining authors to be writers and perform analytics only?

Reviewer #3: Unfortunately, I do not think it is interesting for us to know the ongoing protocol of review studies.

Reviewer #4: The elegibility criteria should be explained more accurately and there is no stratification of patients

7. PLOS authors have the option to publish the peer review history of their article (what does this mean?). If published, this will include your full peer review and any attached files.

Reviewer #1: **Yes: **Dott. Alessandro De Luca

Reviewer #2: No

Reviewer #3: No

Reviewer #4: No

---

## [Author Response · Author response to Decision Letter 0]

29 Nov 2022

PONE-D-22-24531

Comparative efficacy and acceptability of interventions for insomnia in breast cancer patients: A protocol for systematic review and network meta-analysis

PLOS ONE

Dear Dr. Andrea Giannini,

We have received and reviewed the editors' and reviewers’ comments on our manuscript: Comparative efficacy and acceptability of interventions for insomnia in breast cancer patients: A protocol for systematic review and network meta-analysis. Thank you very much for the opportunity to revise our manuscript. A point-by-point response to the comments follows. We highlighted all the changes made to the original manuscript.

My colleagues and I are grateful for your consideration of our work, and we look forward to hearing your decision.

Junqiang Lei

Dear Dr. Jun-Qiang Lei

Thank you for submitting your manuscript to PLOS ONE. After careful consideration, we feel that it has merit but does not fully meet PLOS ONE’s publication criteria as it currently stands. Therefore, we invite you to submit a revised version of the manuscript that addresses the points raised during the review process.

Dear authors,

the topic of the present article titled “Comparative efficacy and acceptability of interventions for insomnia in breast cancer patients: A protocol for systematic review and network meta-analysis” is very interesting, the paper and the aim falls within the scope of the journal but the article needs major improvements.

Review some spellings. Methods and results section and discussion should be modified and improved.

I suggest improving the manuscript with the reviewers' comments

If applicable, we recommend that you deposit your laboratory protocols in protocols.io to enhance the reproducibility of your results. Protocols.io assigns your protocol its own identifier (DOI) so that it can be cited independently in the future. For instructions see:https://journals.plos.org/plosone/s/submission-guidelines#loc-laboratory-protocols. Additionally, PLOS ONE offers an option for publishing peer-reviewed Lab Protocol articles, which describe protocols hosted on protocols.io. Read more information on sharing protocols at https://plos.org/protocols?utm_medium=editorial-email&utm_source=authorletters&utm_campaign=protocols.

Reply: Thank you very much for all these instructions. We revised our manuscript carefully.

Journal Requirements:

Reply: Thank you very much for all these instructions. We revised our manuscript carefully.

Reviewers' comments:

Reviewer's Responses to Questions

Comments to the Author

1. Does the manuscript provide a valid rationale for the proposed study, with clearly identified and justified research questions?

Reviewer #1: Yes

Reviewer #2: Yes

Reviewer #3: No

Reviewer #4: Partly

2. Is the protocol technically sound and planned in a manner that will lead to a meaningful outcome and allow testing the stated hypotheses?

Reviewer #1: Yes

Reviewer #2: Yes

Reviewer #3: Yes

Reviewer #4: Partly

3. Is the methodology feasible and described in sufficient detail to allow the work to be replicable?

Reviewer #1: Yes

Reviewer #2: Yes

Reviewer #3: Yes

Reviewer #4: Yes

4. Have the authors described where all data underlying the findings will be made available when the study is complete?

Reviewer #1: Yes

Reviewer #2: No

Reviewer #3: Yes

Reviewer #4: Yes

5. Is the manuscript presented in an intelligible fashion and written in standard English?

Reviewer #1: Yes

Reviewer #2: Yes

Reviewer #3: Yes

Reviewer #4: Yes

6. Review Comments to the Author

You may also provide optional suggestions and comments to authors that they might find helpful in planning their study.

Reviewer #1: I think the article is very interesting and current. From our knowledge we know that insomnia is determined not only by the diagnosis of the tumor but also by its treatment. it happens very often that after an almost complete recovery, hormone therapy exacerbates a previous insomnia or creates alterations in the sleep wake cycle de novo. I believe it is essential to stress this speech in the introduction and discussion.

Reply: Thank you very much for your positive comments and good suggestion. We have revised the “Introduction” and “Discussion” sections accordingly.

Introduction Section

“For breast cancer patients, in addition to the stress caused by the cancer diagnosis, some treatment-related factors are often related to insomnia. Vasomotor symptoms, including hot sweating and night sweats, are common with chemotherapy-induced amenorrhea and endocrine therapies and can cause insomnia episodes or aggravate pre-existing conditions [11-14].”

Discussion Section

“Approximately 30%-50% of breast cancer survivors have insomnia persisting years after treatment [52]. Stress, anxiety, discomfort after surgery such as chronic pain after breast surgery, and side effects of treatment such as hot sweats and night sweats, which are common after chemotherapy and endocrine therapy, can all contribute to insomnia in breast cancer patients [12, 14, 52, 54, 55].”

Reviewer #2: Question 4 - The authors have stated that all relevant data from the study will be made available upon study completion; this is a protocol, not a completed study.

Reply: Thank you for pointing this out. We have revised this statement.

- The study will explore the "comparative efficacy and acceptability" of interventions for insomnia in breast cancer patients. Efficacy of interventions will be able to be shown once the data is analyzed; however, what is meant by "acceptability" and how will that be explored and analyzed?

Reply: Thank you for pointing this out. We have refined this in the manuscript.

“To measure the acceptability of treatments, we will use the proportion of patients who stopped treatment for any reason (all cause discontinuation) [42].”

“We will use pooled odds ratio (OR) with 95% credible intervals for the dichotomous variables (ie. acceptability).”

- It is surprising that there will be "...no restriction on the language of publication" of studies to be selected. Will there be translation help furnished, or are the authors knowledgable in a large number of languages?

Reply: Thank you for pointing this out. We have added the sentence “For publications that are not in English and Chinese, we will seek translation assistance” to our manuscript.

- The controls will be "...waiting list, treatment/care as usual, etc." Do the authors mean to assemble control patients, find controls outside of the selected articles, or use the "controls" who are presented in the RCTs used?

Reply: Thank you for pointing this out. We will investigate the potential clinical and statistical heterogeneity of non-interventional controls to determine whether to assemble control patients.

“The controls will include waiting list, treatment/care as usual, placebos. We will investigate the potential clinical and statistical heterogeneity of non-interventional controls (waiting list, treatment/care as usual, placebo) to determine whether to treat them as a single node.”

- It is surprising that, out of seven authors, only two will screen all articles and review all the selected articles. Are the remaining authors to be writers and perform analytics only?

Reply: Thank you for pointing this out. We revised “two reviewers” to “pairs of reviewers".

Reviewer #3: Unfortunately, I do not think it is interesting for us to know the ongoing protocol of review studies.

Reply: On behave of the authors, I wish to express my profound gratitude to you for finding time out of your busy schedule to review this manuscript. Our research will explore the comparative effectiveness and acceptability of all currently available interventions for insomnia in patients with breast cancer, and we will use the GRADE approach to assess the certainty of the evidence supporting the effect of treatment. The state-of-the-art methodology will also be used to summarize the relative effectiveness of competing interventions. The results of our review will help provide more evidence for the treatment of insomnia in breast cancer patients.

Reviewer #4: The elegibility criteria should be explained more accurately and there is no stratification of patients

Reply: Thank you for the suggestion. We refined the eligibility criteria and added subgroup analyses based on patient stratification.

“Eligibility criteria

Types of studies

Randomized controlled trials comparing the effects of different interventions on the management of insomnia in breast cancer patients will be included. RCTs should report at least one measure outcome of sleep and provide enough detail to calculate effect sizes. We will exclude cluster or crossover RCTs.

Population

Individuals with a breast cancer diagnosis who had clinically relevant levels of insomnia as defined by a standardized diagnostic system such as the Diagnostic and Statistical Manual of Mental Disorders (DSM) [39], International Classification of Sleep Disorders (ICSD) [40], Tenth revision of the International Statistical Classification of Diseases and Health-Related Problems (ICD-10) [41], and other well-recognized classifications. We will put no restrictions on the subtype, histology, stage, and treatment of breast cancer, age and comorbidities of patients, and duration of insomnia.

Intervention

Any pharmacological and non-pharmacological interventions used for the management of insomnia in breast cancer patients. The eligible interventions are listed in Table 1.

Comparator

The controls will include waiting list, treatment/care as usual, placebo, or a different pharmacological or non-pharmacological intervention. We will investigate the potential clinical and statistical heterogeneity of non-interventional controls (waiting list, treatment/care as usual, placebo) to determine whether to treat them as a single node.

Outcome

Outcomes of interest are sleep efficiency (SE, %), total sleep time (TST), sleep onset latency (SOL), wake after sleep onset (WASO), acceptability. SE, SOL, WASO, and TST are typically defined as a ratio of time asleep to time in bed, the length of time after lights out until sleep onset, the length of waking time after the onset of persistent sleep, and total nighttime sleep, respectively. In addition, the scores from the Insomnia Severity Index (ISI) or the Pittsburgh Sleep Quality Index (PSQI) are also the outcomes that we are interested in. For both questionnaires, a higher score indicates more severe insomnia (ISI) or worse sleep quality (PSQI). To measure the acceptability of treatments, we will use the proportion of patients who stopped treatment for any reason (all cause discontinuation) [42].”

“Subgroup analysis and sensitivity analysis

Where sufficient data are available, we will conduct subgroup analyses or meta-regression analyses based on country and mean age of patients; subtype, histology, stage, and treatment (eg, surgery, chemotherapy, endocrine therapy) of breast cancer; comorbid disorders, and duration of interventions and insomnia. We will also conduct sensitivity analyses to explore the influence of variables on the outcomes. Planned sensitivity analyses will exclude trials with a high risk of bias or significant levels of missing data.”

7. PLOS authors have the option to publish the peer review history of their article (what does this mean?). If published, this will include your full peer review and any attached files.

Do you want your identity to be public for this peer review? For information about this choice, including consent withdrawal, please see our Privacy Policy.

Reviewer #1: Yes: Dott. Alessandro De Luca

Reviewer #2: No

Reviewer #3: No

Reviewer #4: No

---

## [Decision Letter · Decision Letter 1]

6 Jan 2023

PONE-D-22-24531R1Comparative efficacy and acceptability of interventions for insomnia in breast cancer patients: A protocol for systematic review and network meta-analysisPLOS ONE

Dear Dr. Junqiang,

Thank you for submitting your manuscript to PLOS ONE. After careful consideration, we feel that it has merit but does not fully meet PLOS ONE’s publication criteria as it currently stands. Therefore, we invite you to submit a revised version of the manuscript that addresses the points raised during the review process.

We look forward to receiving your revised manuscript.

Kind regards,

Andrea Giannini

Academic Editor

PLOS ONE

Journal Requirements:

Additional Editor Comments:

Dear authors,

the manuscript it has now been evaluated by our experts and they have recommended that minor changes be made to the submission.

Please improving the manuscript with the reviewers' comments.

Reviewers' comments:

Reviewer's Responses to Questions

**Comments to the Author**

1. Does the manuscript provide a valid rationale for the proposed study, with clearly identified and justified research questions?

Reviewer #1: Yes

Reviewer #2: Yes

Reviewer #3: Yes

2. Is the protocol technically sound and planned in a manner that will lead to a meaningful outcome and allow testing the stated hypotheses?

Reviewer #1: Yes

Reviewer #2: Yes

Reviewer #3: Partly

3. Is the methodology feasible and described in sufficient detail to allow the work to be replicable?

Reviewer #1: Yes

Reviewer #2: Yes

Reviewer #3: Yes

4. Have the authors described where all data underlying the findings will be made available when the study is complete?

Reviewer #1: Yes

Reviewer #2: Yes

Reviewer #3: Yes

5. Is the manuscript presented in an intelligible fashion and written in standard English?

Reviewer #1: Yes

Reviewer #2: Yes

Reviewer #3: Yes

6. Review Comments to the Author

You may also provide optional suggestions and comments to authors that they might find helpful in planning their study.

Reviewer #1: I think the article is very interesting and current.It is very important for clinical practice to investigate issues such as insomnia in women diagnosed with breast cancer. This study certainly addresses an issue that needs more study and therefore more attention.

Reviewer #2: - Although the authors did answer all the questions from the reviewers and stated how they changed the text, they did not provide a version of the manuscript with tracked changes. Thus, it was extremely difficult to find where the text had been changed; reviewers could only see what the rewrite was by itself.

- In several places, the term "hot sweating" was used as one of the vasomotor symptoms. I believe that this is used instead of "hot flashes" or "hot flushes"? This should be changed.

- It is still not very clear what "acceptability" means; I believe it is meant how well the intervention was accepted by patients and whether they continued to use the intervention because it was effective. Could this be clarified?

Reviewer #3: The results of this study will attract physician's interest. I hope this study will provide some evidence for the management of the insomnia in breast cancer patients.

7. PLOS authors have the option to publish the peer review history of their article (what does this mean?). If published, this will include your full peer review and any attached files.

Reviewer #1: No

Reviewer #2: No

Reviewer #3: No

---

## [Author Response · Author response to Decision Letter 1]

9 Feb 2023

PONE-D-22-24531R1

Comparative efficacy and acceptability of interventions for insomnia in breast cancer patients: A protocol for systematic review and network meta-analysis

PLOS ONE

Dear Dr. Andrea Giannini,

We have received and reviewed the editors' and reviewers’ comments on our manuscript: Comparative efficacy and acceptability of interventions for insomnia in breast cancer patients: A protocol for systematic review and network meta-analysis. Thank you very much for the opportunity to revise our manuscript. A point-by-point response to the comments follows. We highlighted all the changes made to the original manuscript.

My colleagues and I are grateful for your consideration of our work, and we look forward to hearing your decision.

Junqiang Lei

Dear Dr. Junqiang,

Thank you for submitting your manuscript to PLOS ONE. After careful consideration, we feel that it has merit but does not fully meet PLOS ONE’s publication criteria as it currently stands. Therefore, we invite you to submit a revised version of the manuscript that addresses the points raised during the review process.

Reply: Thank you very much for all these instructions. We revised our manuscript carefully.

Journal Requirements:

Additional Editor Comments:

Dear authors,

the manuscript it has now been evaluated by our experts and they have recommended that minor changes be made to the submission.

Please improving the manuscript with the reviewers' comments.

Reply: Thank you very much for all these instructions. We revised our manuscript carefully.

Reviewers' comments:

Reviewer's Responses to Questions

Comments to the Author

1. Does the manuscript provide a valid rationale for the proposed study, with clearly identified and justified research questions?

Reviewer #1: Yes

Reviewer #2: Yes

Reviewer #3: Yes

2. Is the protocol technically sound and planned in a manner that will lead to a meaningful outcome and allow testing the stated hypotheses?

Reviewer #1: Yes

Reviewer #2: Yes

Reviewer #3: Partly

3. Is the methodology feasible and described in sufficient detail to allow the work to be replicable?

Reviewer #1: Yes

Reviewer #2: Yes

Reviewer #3: Yes

4. Have the authors described where all data underlying the findings will be made available when the study is complete?

Reviewer #1: Yes

Reviewer #2: Yes

Reviewer #3: Yes

5. Is the manuscript presented in an intelligible fashion and written in standard English?

Reviewer #1: Yes

Reviewer #2: Yes

Reviewer #3: Yes

6. Review Comments to the Author

You may also provide optional suggestions and comments to authors that they might find helpful in planning their study.

Reviewer #1: I think the article is very interesting and current. It is very important for clinical practice to investigate issues such as insomnia in women diagnosed with breast cancer. This study certainly addresses an issue that needs more study and therefore more attention.

Reply: Thank you very much for your positive comments.

Reviewer #2: - Although the authors did answer all the questions from the reviewers and stated how they changed the text, they did not provide a version of the manuscript with tracked changes. Thus, it was extremely difficult to find where the text had been changed; reviewers could only see what the rewrite was by itself.

Reply: Thank you very much for your comments. We have provided two versions of manuscript: a version of manuscript with marked changes; an unmarked version of our revised paper without tracked changes.

- In several places, the term "hot sweating" was used as one of the vasomotor symptoms. I believe that this is used instead of "hot flashes" or "hot flushes"? This should be changed.

Reply: Thank you for pointing this out. We have changed "hot sweating" to "hot flashes" across our manuscript.

- It is still not very clear what "acceptability" means; I believe it is meant how well the intervention was accepted by patients and whether they continued to use the intervention because it was effective. Could this be clarified?

Reply: Thank you for your good suggestion. We revised our manuscript accordingly.

“To measure the acceptability of treatments, we will use the proportion of patients who stopped treatment for any reason (all-cause discontinuation). Acceptability reflects how well the intervention was accepted by patients and whether they continued to use the intervention because it was effective.”

Reviewer #3: The results of this study will attract physician's interest. I hope this study will provide some evidence for the management of the insomnia in breast cancer patients.

Reply: Thank you very much for your interest in our study.

7. PLOS authors have the option to publish the peer review history of their article (what does this mean?). If published, this will include your full peer review and any attached files.

Do you want your identity to be public for this peer review? For information about this choice, including consent withdrawal, please see our Privacy Policy.

Reviewer #1: No

Reviewer #2: No

Reviewer #3: No

---

## [Decision Letter · Decision Letter 2]

20 Feb 2023

Comparative efficacy and acceptability of interventions for insomnia in breast cancer patients: A protocol for systematic review and network meta-analysis

PONE-D-22-24531R2

Dear Dr. Junqiang,

We’re pleased to inform you that your manuscript has been judged scientifically suitable for publication and will be formally accepted for publication once it meets all outstanding technical requirements.

Kind regards,

Andrea Giannini

Academic Editor

PLOS ONE

Additional Editor Comments (optional):

The manuscript has been modified with the comments of the reviewers. It is now ready to be published.

Reviewers' comments:

Reviewer's Responses to Questions

**Comments to the Author**

1. Does the manuscript provide a valid rationale for the proposed study, with clearly identified and justified research questions?

Reviewer #1: Yes

Reviewer #2: Yes

2. Is the protocol technically sound and planned in a manner that will lead to a meaningful outcome and allow testing the stated hypotheses?

Reviewer #1: Yes

Reviewer #2: Yes

3. Is the methodology feasible and described in sufficient detail to allow the work to be replicable?

Reviewer #1: Yes

Reviewer #2: Yes

4. Have the authors described where all data underlying the findings will be made available when the study is complete?

Reviewer #1: Yes

Reviewer #2: Yes

5. Is the manuscript presented in an intelligible fashion and written in standard English?

Reviewer #1: Yes

Reviewer #2: Yes

6. Review Comments to the Author

You may also provide optional suggestions and comments to authors that they might find helpful in planning their study.

Reviewer #1: dear author, I really appreciated the study and the topic addressed.

This is a very interesting and original article. QoL remains a very important aspect both at diagnosis and during treatment for breast cancer. Good work

Reviewer #2: The authors have provided answers and revisions that were commented upon previously to my satisfaction.

7. PLOS authors have the option to publish the peer review history of their article (what does this mean?). If published, this will include your full peer review and any attached files.

Reviewer #1: No

Reviewer #2: No

---

## [Editor Report · Acceptance letter]

24 Feb 2023

PONE-D-22-24531R2 

Comparative efficacy and acceptability of interventions for insomnia in breast cancer patients: A protocol for systematic review and network meta-analysis 

Dear Dr. Lei:

I'm pleased to inform you that your manuscript has been deemed suitable for publication in PLOS ONE. Congratulations! Your manuscript is now with our production department. 

Kind regards, 

on behalf of

Dr. Andrea Giannini 

Academic Editor

PLOS ONE